# Comparison of Localization Methods in Cushing Disease—Could [^11^C]C-Methionine PET/CT Replace MRI or BIPSS?

**DOI:** 10.3390/cancers17193147

**Published:** 2025-09-27

**Authors:** Adam Daniel Durma, Marek Saracyn, Maciej Kołodziej, Grzegorz Zieliński, Piotr Zięcina, Jerzy Narloch, Grzegorz Kamiński

**Affiliations:** 1Department of Endocrinology and Radioisotope Therapy, Military Institute of Medicine—National Research Institute, 04-141 Warsaw, Poland; 2Faculty of Medicine, Warsaw University, 02-089 Warsaw, Poland; 3Department of Neurosurgery, Military Institute of Medicine—National Research Institute, 04-141 Warsaw, Poland; 4Department of Interventional Radiology, Military Institute of Medicine—National Research Institute, 04-141 Warsaw, Poland

**Keywords:** [^11^C]C-methionine, [^11^C]C-MET, PET/CT, bilateral inferior petrosal sinus sampling, Cushing disease

## Abstract

**Simple Summary:**

This prospective study assessed whether [^11^C]C-MET PET/CT could serve as a non-invasive alternative to BIPSS and MRI in diagnosing Cushing disease. [^11^C]C-MET PET/CT showed limited sensitivity (24%) and specificity (13%) for detecting corticotrope pituitary adenomas comparing to other methods. However, additional analysis revealed significantly higher SUV_max_ values in patients with corticotrope adenomas compared to control group. BIPSS remains the most reliable diagnostic method for CD; however, [^11^C]C-MET uptake patterns may reflect increased amino acid metabolism, suggesting possible adenoma presence.

**Abstract:**

**Introduction**: Cushing syndrome (CS) is a medical condition resulting from prolonged hypercortisolemia. The most common reason for endogenous CS is ACTH overproduction by pituitary adenoma, and then it is called Cushing disease (CD). The gold standard of CD diagnostic remains bilateral inferior petrosal sinus sampling (BIPSS); nevertheless, non-invasive diagnostic methods are being sought to provide a higher safety profile. The aim of this study was to evaluate whether [^11^C]C-MET PET/CT can serve as a non-invasive alternative to BIPSS and MRI in CD diagnosis. **Methods**: This prospective study included 21 patients with CD who underwent BIPSS, MRI of the pituitary, and [^11^C]C-MET PET/CT. **Results**: Sensitivity of BIPSS, MRI and [^11^C]C-MET PET/CT was 100%, 59% and 24%, respectively, while specificity was 100%, 75%, and 13%. Next, we retrospectively compared PET/CT results for patients with corticotrope pituitary adenomas (*n* = 18) with those for individuals with no pituitary pathology (*n* = 18), and the results showed significantly higher SUV_max_ in the study group (3.74 ± 0.90 vs. 1.87 ± 1.17; *p* < 0.001). In ROC curve analysis, the area under the curve (AUC) was 0.889 (*p* <0.001; 95% CI 0.784–0.994). For SUV_max_ 2.60, the calculated sensitivity and specificity were 89% and 78% respectively, and for SUV_max_ 3.56, sensitivity and specificity were 67% and 89%, respectively. **Conclusions**: [^11^C]C-MET PET/CT seems not to be a reliable diagnostic option in the diagnosis of pituitary corticotropic adenomas. BIPSS proved still to be the best diagnostic option for CD. Nevertheless, a higher than normal pituitary accumulation of the radiotracer may suggest the presence of increased amino acid metabolism, thus, the presence of adenoma.

## 1. Introduction

Cushing syndrome (CS) is a condition caused by prolonged hypercortisolemia, characterized by symptoms such as hypertension, central obesity, red stretch marks, a moon-shaped face, facial redness, a fat lump on the shoulders, acne, fatigue, glucose metabolism disturbances, and irregular menstruation in females [1]. CS can be classified as either exogenous, resulting from chronic corticosteroid use, or endogenous, most commonly caused by a corticotrope pituitary adenoma [1,2]. Endogenous ACTH-dependent CS is referred to as Cushing disease (CD).

Diagnosing CD is complex and often challenging. Laboratory results can be ambiguous, and standard imaging techniques may fail to detect a pituitary adenoma or other cause of hormone overproduction [3]. The initial screening tests for CS include late-night salivary cortisol (LNSC), 24-h urine free cortisol (UFC), and the overnight dexamethasone suppression test (ODST) [4]. Positive results of these tests warrant referral to a specialized center for further diagnostics. To confirm hypercortisolemia, a low-dose dexamethasone suppression test (LDDST) is recommended, along with ACTH concentration measurements to determine the source of cortisol excess [5,6]. In suspicion of ectopic CS, a high-dose dexamethasone suppression test (HDDST) should also be considered [7].

Imaging for CS diagnosis typically includes magnetic resonance imaging (MRI) of the pituitary and computed tomography (CT) of the adrenal glands [8,9,10]. MRI does not always confirm the presence of adenoma; up to 40% of cases are “MRI–negative” or “MRI-inconclusive”. In such cases, the next diagnostic step should be bilateral inferior petrosal sinus sampling (BIPSS), performed in order to confirm the diagnosis and guide treatment planning.

BIPSS is a complex intravascular procedure that involves inserting catheters through the femoral vein into the inferior petrosal sinuses [11,12]. Blood samples are collected from the left and right inferior petrosal sinuses, as well as from the peripheral site, at intervals of −1, 0, 1, 3, 5, and 10 min following intravenous administration of 10 µg of desmopressin or 100 µg of corticotropin-releasing hormone (CRH). This test is crucial to differentiate between central and ectopic sources of ACTH-dependent hypercortisolemia and can provide insights into the lateralization of pituitary adenomas. BIPSS requires a skilled team of interventional radiologists, as it carries risks such as thromboembolism and vascular rupture. Additionally, the procedure involves significant exposure to iodine contrast and ionizing radiation, as it is performed under X-ray guidance.

Positron emission tomography (PET) is an imaging technique that uses isotopes undergoing beta-plus (β^+^) decay. Certain isotopes, such as fluorine-18 (^18^F) and gallium-68 (^68^Ga), emit positrons—positively charged particles with the same mass as electrons—during decay [13]. When a positron collides with an electron, an annihilation process occurs, producing gamma (γ) photons, which are detected by the PET scanner [14]. PET enables non-invasive lesion localization based on metabolic activity or the presence of specific receptors. A low-dose CT scan is often performed simultaneously to enhance image quality, correct for tissue attenuation, and provide anatomical context for regions of high radiotracer accumulation, creating a PET/CT fusion image. PET can also be combined with MRI to form a PET/MRI scan [15,16,17,18].

For many years, there has been a search for new radiotracers that could be used for non-invasive diagnosis of central nervous system tumors, including pituitary tumors [19]. One of the most promising radiotracers in such cases seemed to be methionine, especially one labeled with β^+^-emitting carbon-11. This [^11^C]C-methionine (or simply [^11^C]C-MET) has shown utility in many previous studies [20,21,22,23].

Methionine itself is an essential amino acid (AA) with a five-carbon chain and a distinctive sulfur atom in its structure. In eukaryotic cells, methionine serves as the universal start AA for protein synthesis. Due to its role in protein synthesis and cellular energy production, methionine has been used as a carrier for radioisotopes in the detection of tumors with increased AA metabolism. The diagnostic value of [^11^C]C-MET PET/CT has been particularly well established in neuro-oncology for detecting gliomas and myelomas, especially in borderline cases [24,25,26,27]. Additionally, studies on endocrine disorders and neoplasms have highlighted the method’s high sensitivity and accuracy [28,29,30].

An alternative label for [^11^C]C-methionine is [^13^N]N-ammonia ([^13^N]N-NH_3_). In a study by Wang et al., both markers were compared in a group of nine patients, with a higher uptake of [^11^C]C-MET observed in pituitary adenomas (PAs) compared to pituitary tissues, while [^13^N]N-NH_3_ had a higher uptake in pituitary tissues than in pituitary adenomas [31]. Combining those radiotracers has proven to be effective in the differentiation of PAs from pituitary tissue in recurrent functional PAs, especially with negative MRI or [^18^F]F-FDG PET studies.

Other diagnostic options include [^68^Ga]Ga-DOTA-TATE, which has an affinity for somatostatin receptors present on most neuroendocrine tumor cells, including corticotropinomas (currently classified as PiTNEN); [^18^F]fluoroethyl-L-tyrosine ([^18^F]FET), which becomes incorporated into cellular metabolism; or even the newest of the radiotracers—[^68^Ga]Ga-DOTA-mDesmo, which targets V1b receptors specifically overexpressed in CD [32,33,34]. Some of these tests may even be used for differential diagnosis of central versus ectopic origin of Cushing’s disease [35]. The limitations of most of the studies available so far are their retrospective nature and relatively small control groups; however, the need to find alternative and non-invasive methods for the standard ones (BIPSS) encourages the intensification of ongoing research.

The aim of this study was to prospectively evaluate whether [^11^C]C-MET PET/CT can serve as a non-invasive alternative to BIPSS, providing comparable sensitivity and specificity.

## 2. Materials and Methods

This prospective study included 21 patients with laboratory-confirmed ACTH-dependent CS who underwent extensive diagnostic evaluation at the highest reference center. On Day 1, patients underwent [^11^C]C-MET PET/CT, followed by BIPSS with 10 µg of desmopresin on Day 2, and pituitary MRI on Day 3. If CD was confirmed, patients underwent transsphenoidal surgery within the next 2–4 weeks, followed by histological evaluation of the resected adenomas.

For a more detailed assessment of PET/CT results and parameters, we compared the study group to a control group of patients who underwent [^11^C]C-MET PET/CT in a different study performed in our Center [36]. This retrospectively added control group consisted of 18 patients with tertiary hyperparathyroidism who had functional pituitary adenomas excluded by clinical examination and laboratory results.

### 2.1. Study Inclusion Criteria

ACTH-dependent Cushing syndromeAge over 18.Signing informed consent to participate in the study.

### 2.2. Study Exclusion Criteria

Pregnancy or lactation.Previous pituitary surgery.Allergy to gadolinium contrast.Lack of signing informed consent.Age below 18.

### 2.3. PET Protocol

PET/CT studies were conducted at the Mazovian PET/CT Center “Affidea” using a 64-row Discovery 710 scanner (GE Medical Systems, Milwaukee, WI, USA). Acquisition was performed from the skull base to the clavicles, starting 20 min after intravenous administration of approximately 500 MBq (±10%) of [11C]C-MET, with a total scan duration of 20 min. PET/CT images were independently reviewed by two external, board-certified nuclear medicine specialists (each with >15 years of experience, not involved in the study, blinded to other clinical results) and subsequently confirmed by a third nuclear medicine physician (study co-author, unblinded to other results). Image evaluation was performed using the multimodal Volume Share 5–Advantage Workstation 4.6 (GE Medical Systems).

For the CT acquisition, the following parameters were applied: tube voltage 140 kV; tube current automatically adjusted to patient body weight, 40–100 mA; noise index 22; slice thickness for attenuation correction 3.75 mm (reconstructed to 1.25 mm); gantry rotation time 0.8 s; and pitch 0.984:1.

For the PET acquisition, iterative reconstruction with Vue Point Fx was applied (18 subsets, 3 iterations, matrix size 256 × 256). In all examinations, the Time-of-Flight (ToF) technique was employed to improve diagnostic confidence and enhance image contrast relative to noise.

### 2.4. Methionine Protocol

[^11^C]C-MET ([S-methyl-^11^C]-L-methionine) was synthesized in Synektik Research Center, Warsaw, Poland due to European Pharmacopoeia regulations. For synthesis gas, nitrogen was irritated in a cyclotron by proton bombardment, and the reaction can be written as 14N(p,α)11C.

During synthesis, key parameters (activity, temperature, reagents) were controlled using the Modular-Lab unit. Cyclotron-produced [^11^C]CO_2_ was trapped in the CTM, released by heating to 400 °C under helium flow, and transferred to cooled reaction vessels. It was reduced to [^11^C]CH_3_OH with lithium aluminum hydride in THF at 100 °C, followed by solvent evaporation at 150 °C. Subsequent reaction with 58% HI generated [^11^C]CH_3_I, which was passed through NaOH and into an HLB cartridge containing the precursor (L-homocysteine thiolactone hydrochloride). A spontaneous room-temperature reaction occurred, monitored until maximum activity was reached. The final product was eluted with 8 mL sodium dihydrogen phosphate buffer.

[^11^C]C-MET synthesis efficiency was 34.37 ± 12.31%; D,L-[^11^C]C-MET radiochemical purity was 97.72 ± 1.99%; and proper activity was 6.52 ± 1.56 GBq/mg.

### 2.5. BIPSS Protocol

The bilateral inferior petrosal sinus sampling (BIPSS) examination was performed in the Interventional Radiology Department of the Military Institute of Medicine—National Research Institute using the Innova IGS6 (General Electric, Milwauke, WI, USA). The study used a non-ionic, iso-osmolar contrast medium Visipaque 320 (General Electric, USA) using manual injections of 8–10 mL to each vessel. In the first step, bilateral access to the common femoral veins was obtained using the Seldinger method under local anesthesia. An introducing sheath (6F, 11 cm, Arrow International Inc., Morrisville, NC, USA) was placed with systemic intravenous heparinization using 50 IU of unfractionated heparin per kg of body weight, max. 5000 IU in a bolus. Then, using a coaxial catheter and guidewire system (Head Hunter 5F and a 0.035 hydrophilic guidewire, Merit Medical, Sout Jordan, UT, USA), both internal jugular veins were selectively catheterized, and their venography was performed with visualization of the origin of the inferior petrosal sinuses (IPS), using oblique RAO/LAO 30° projections. After visualization of the origin of IPS, using road map, the tips of the catheters were placed between the horizontal and vertical part of the IPS. Selective venography was performed in the anteroposterior and oblique projections. A necessary condition for confirming the correct position of the catheter was to obtain an image of the contralateral IPS after administration of contrast media to the ipsilateral IPS. After obtaining the correct bilateral position of the catheters, blood samples of approximately 2 mL were taken simultaneously from both IPS and the peripheral puncture (introducer sheath in right femoral vein) at −1 and 0 min before and 1, 3, 5, 10 and 15 min after intravenous administration of 10 μg of desmopressin to determine ACTH concentrations. The correct positions of the catheters in both IPS were checked under road map control between subsequent blood sample collections. Afterward, the catheterization procedure was completed. The condition of the intracranial venous system was checked (control examination—sinusography with venography). The puncture sites of the veins were blocked in the groins, and pressure dressings were applied for 2 h. A bed rest regimen was recommended for 12 h. The following recommendations were used to prevent thromboembolic complications: (1) antithrombotic stockings put on 1 h before the catheterization procedure, (2) low molecular weight heparins (Eonoxaparin, Clexane) at a dose of 40 mg in subcutaneous injections once daily, (3) early patient mobilization. The absorbed dose of X-rays during the examination was 200–300 mGy.

### 2.6. MRI

MRI was performed with use of a Discovery MR750W3T 3.0 Tesla (GE Healthcare, Florence, SC, USA). A standard protocol for pituitary MRI was used, including sagittal and coronal T1 and T2 sequences, with slice thickness of 2 mm. MRI studies were assessed by an external radiologist not involved in the study.

### 2.7. Statistical Analysis

The statistical analysis was performed using IBM SPSS (version 26). To verify whether the results met the rules of normal distribution, the Shapiro–Wilk test was conducted. Results with normal distribution were presented as means (M) and standard deviations (SD), and in the case of non-normal distribution, as medians (Med.) and interquartile ranges (IQR). Differences between groups were analyzed using appropriate tests, such as Student’s *t*-test and U Mann–Whitney test. The Levene test was used to assess the equality of variances in analyzed groups. Pearson or Spearman tests were used for correlation analysis (for normally distributed and not normally distributed data, respectively). ROC curves were used for assessment of the sensitivity and specificity of the PET/CT study. A significance level of *p* < 0.05 was adopted.

## 3. Results

Analysis showed that MRI studies for the detection of pituitary corticotropic adenoma were positive in 11/21 patients (one macroadenoma, and 10 microadenomas), with median (IQR) adenoma size of 4 (3) mm. However, the MRI studies resulted in 10 true-positives (TP), one false-positive (FP), three true-negatives (TN), and seven false-negatives (FN).

The [^11^C]C-MET PET/CT was positive in 10/21 cases and negative in 11/21 cases. In three cases, it was true-positive, while seven cases were false-positives. It was true-negative in one case and false-negative in 10 cases. The images presenting TP, FP, TN, and FN of [^11^C]C-MET PET/CT studies can be found in Figure 1, Figure 2, Figure 3 and Figure 4 and Appendix A.

BIPSS was positive in 18/21 patients, being true-positive in 18. Three remaining cases were confirmed to be ectopic CS (true-negative).

Calculated sensitivity, specificity, positive predictive value (PPV) and negative predictive value (NPV) are presented in Table 1 below.

Of 21 patients from the study group, 18 were finally confirmed to have ACTH-dependent Cushing disease (pituitary adenoma) and underwent transsphenoidal surgery. Seventeen patients had microadenoma (range of 3–9 mm), while one had macroadenoma (13 mm). The remaining three had ACTH-dependent ectopic CS, confirmed in further diagnostic procedures by other modalities. Thus, for analysis regarding PET/CT parameters, we included only confirmed CD patients—the “final study group” (18 patients)—and compared them with 18 patients with normal image and function of the pituitary gland—the “control group”.

In the “final” study group (*n* = 18), 16 patients were female, and two were male. In the control group (*n* = 18) nine were male and nine were female. Detailed data of the study group are presented in Table 2. In the control group, mean age was 46.7 ± 14.1 years old (y.o.).

The mean maximum standardized uptake value (SUV_max_) of the pituitary gland in the study group was significantly higher than in the control group (3.74 ± 0.90 vs. 1.87 ± 1.17; *p* < 0.001); however, analysis of lateralization showed that on the side of the tumor, the activity accumulation was insignificantly lower (0.97×).

In the hindsight analysis, the SUV_max_ of the pituitary in the study group (corticotropic adenoma confirmed) ranged from 2.02 to 5.38, and in the control group, only 0.44 to 3.81. Thus, an ROC curve was prepared to analyze potential cutoffs for sensitivity and specificity in detecting hyperfunction/adenoma. The area under the curve (AUC) was 0.889 (*p* <0.001; 95% CI 0.784–0.994). For SUV_max_ 2.60, the calculated sensitivity and specificity were 89% and 78%, respectively. For SUV_max_ 3.56, the calculated sensitivity and specificity were 67% and 89%, respectively (Figure 5).

In correlation analysis, in the study group only, radiotracer activity was positively corelated with SUV_max_ (*p* = 0.032; R = 0.506; β = 0.011). Neither ACTH, cortisol concentrations, nor changes in ACTH in the CRH test (during BIPSS) were statistically significant (*p* = 0.082; 0.213; 0.174; 0.858; 0.682, respectively).

## 4. Discussion

To the best of our knowledge, this was the first prospective study evaluating the usefulness of [^11^C]C-MET PET/CT in the diagnosis of Cushing disease. It aimed to determine whether the method could accurately detect pituitary corticotrope adenomas and achieve clinical results comparable to those with BIPSS. Regrettably, similar sensitivity and specificity proved to be unattainable. The significant difference in the quality of the examination using [^11^C]C-MET compared not only to BIPSS but also to MRI does not currently allow it to be recommended as a diagnostic alternative and also discourages further research—at least in combination with the radiotracer with PET/CT examination and the use of the current examination protocol.

However, it is noteworthy that the SUV_max_ value was significantly higher in patients with a final diagnosis of corticotrope adenoma compared to the group of patients with a normal pituitary gland. This gives hope for the use of the radiotracer in other protocols and in combination with other modalities like high-resolution MRI.

We also found no significant correlations between laboratory parameters and SUV_max_, and no significant differences were found in assessing the lateralization of radiotracer accumulation in the pituitary gland with corticotrope adenoma.

These unsatisfactory results are in contrast with the majority of previous studies available in the literature. One by Furnica et al. presents data of 22 patients with residual or relapsing corticotrope adenoma. Positive [^11^C]C-MET PET/CT had a detection rate accuracy of 86%, proving reliability of the method in this specific group of patients [30].

In another retrospective dual-center cohort study including 15 patients who had CD confirmed biochemically and histologically (transsphenoidal surgery), six patients underwent either [^11^C]C-MET PET/MRI or 18-fluoro-ethyl-L-thyrozine ([^18^F]FET) PET/MRI [37]. Moreover, three individuals underwent both scans. In the study, MRI detected 67% of the tumors, while [^11^C]C-MET detected only 56%, and [^18^F]FET 78%. All tumors were microadenomas. Interestingly, in one case, [^11^C]C-MET PET/MRI identified a tumor on the opposite side of the expected location, leading to an overall sensitivity and specificity for tumor localization of 89% (95% CI: 51.7–99.7%).

In a study by Feng et al., 15 patients with CD underwent [^18^F]FDG PET/CT and [^11^C]C-MET PET/CT [38]. On [^18^F]FDG PET/CT, eight out of 15 patients (53%) exhibited positive findings, all of which were confirmed as true positives. All 15 patients (100%) diagnosed with CD demonstrated positive results on [^11^C]C-MET PET/CT, giving high expectation to the study.

In another retrospective study by Ikeda et al., analysis encompassed 35 patients diagnosed with CD (confirmed through surgical pituitary exploration) [39]. Patients underwent superconductive MRI (1.5 or 3.0 T) alongside composite imaging from [^18^F]FDG PET or [^11^C]C-MET PET combined with 3.0-T MRI to compare adenoma localization with surgical findings. Superconductive MRI exhibited a diagnostic accuracy of only 40% for detecting CD adenomas, with false-negative results in 10 cases and false-positive findings in six cases, with three cases involving double pituitary adenomas. Conversely, the localization accuracy for microadenomas using [^11^C]C-MET PET/MRI was 100%, while [^18^F]FDG PET/MRI demonstrated an accuracy of 73%. No significant differences were observed in the SUV_max_ values for adenomas assessed by [^11^C]C-MET PET between preclinical and overt CD cases.

In a retrospective study by Ishida et al. involving 15 patients with recurrent CD, assessment of the utility of [^11^C]C-MET PET/CT was performed [40]. Due to ambiguous MRI findings, they assumed increased radiotracer accumulation may suggest tumor remnants. All participants had previously undergone transsphenoidal surgery, and MRI scans revealed poorly enhanced lesions that were difficult to distinguish from postoperative alterations. Positive [^11^C]C-MET uptake was detected in eight patients, whereas seven patients exhibited negative uptake. Corticotrope tumors were identified in five cases, including one with negative [^11^C]C-MET uptake. Notably, in two patients [^11^C]C-MET uptake suggested tumor locations contrary to those suggested by MRI.

In a study by Koulouri et al. involving 10 patients diagnosed with de novo CD, eight with persistent or recurrent hypercortisolism following initial pituitary surgery and two with ectopic CS, all individuals underwent [^11^C]C-MET PET/CT [41]. Among the 10 de novo CD cases, seven displayed asymmetric [^11^C]C-MET uptake within the pituitary gland, aligning with suspected corticotrope microadenomas observed in MRI. Subsequent transsphenoidal surgery confirmed these findings histologically. Focal [^11^C]C-MET uptake, corresponding to abnormalities suspected on MRI, was identified in five of the eight patients with recurrent hypercortisolism. In the two cases of ectopic CS, [^11^C]C-MET uptake was predominantly observed in distant metastatic sites with minimal uptake in the sellar region, finally confirming the presence of neoplastic foci.

Nevertheless, studies with [^11^C]C-MET are corelated with a number of potential pitfalls, as radiotracer can present a wide spectrum of accumulation in different intracranial lesions; thus, interpretation of [^11^C]C-MET images requires a high degree of experience [42,43].

The studies mentioned above show the possible use of [^11^C]C-MET; however, their results differ significantly, from those indicating almost 100% sensitivity to those indicating approximately 50% sensitivity, to false positives and ambiguous findings. The use of MRI instead of CT as an anatomical scan seems to significantly improve the quality of the research. Despite the similar quality of the protocols, such large differences may result from differences in populations, different mutations causing adenoma formation, or different image or data analysis. Moreover, the reason may lie in the fact that the pituitary gland physiologically accumulates methionine (and therefore its ^11^C-isotope), which may limit the sensitivity of the study in some patients. In the studies presented in the discussion, the sensitivity and specificity of the test were overestimated in postoperative cases, where pathological accumulation of the radiotracer was visible in tissues that physiologically do not accumulate such a marker.

## 5. Study Weaknesses and Strengths

The study group was relatively small; however, we have used for the first time in the literature a prospective protocol, and initial unsatisfying results did not allow for project continuation. Nevertheless, the study was a prospective one, and all patients underwent a complete set of imaging studies, proper diagnosis, treatment and follow-up.

Despite using the prospective protocol, for now, we cannot recommend the [^11^C]C-MET PET/CT as a reliable, non-invasive replacement for BIPSS study. Literature analysis points to some specific groups of patients in which [^11^C]C-MET PET might be useful (like recurrent/residual/ectopic disease); nevertheless, further prospective studies are necessary to evaluate our observations and the possible use of [^11^C]C-MET in CD diagnosis.

## 6. Conclusions

[^11^C]C-MET PET/CT seems not to be a valuable diagnostic option in the diagnosis of pituitary corticotropic adenomas. BIPSS still proves to be a “gold standard” in the diagnosis of CD. Nevertheless, a higher than normal pituitary accumulation of the radiotracer may suggest the presence of increased amino acid metabolism, thus, the presence of adenoma.

## Figures and Tables

**Figure 1 cancers-17-03147-f001:**
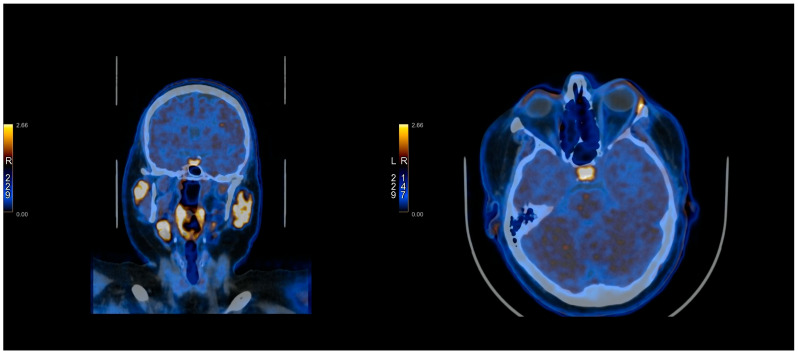
[^11^C]C-MET PET/CT study—True positive (Patient No. 1).

**Figure 2 cancers-17-03147-f002:**
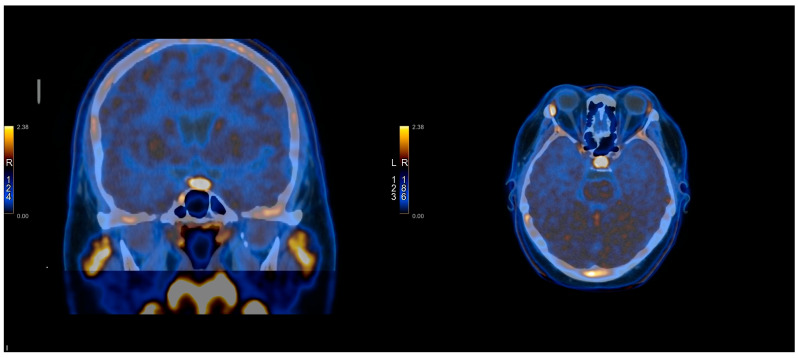
[^11^C]C-MET PET/CT study—True negative (Patient No. 8).

**Figure 3 cancers-17-03147-f003:**
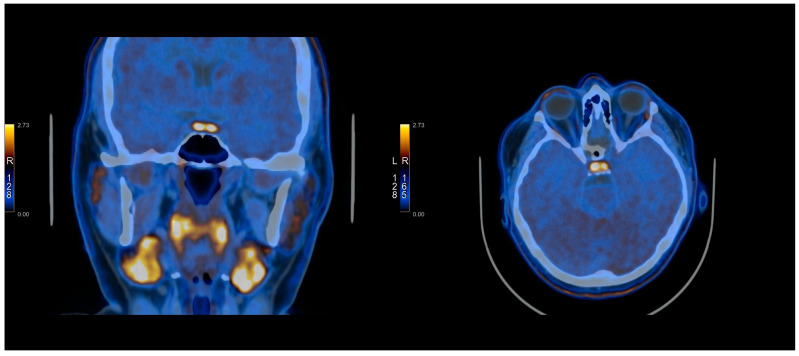
[^11^C]C-MET PET/CT study—False positive (Patient No. 12).

**Figure 4 cancers-17-03147-f004:**
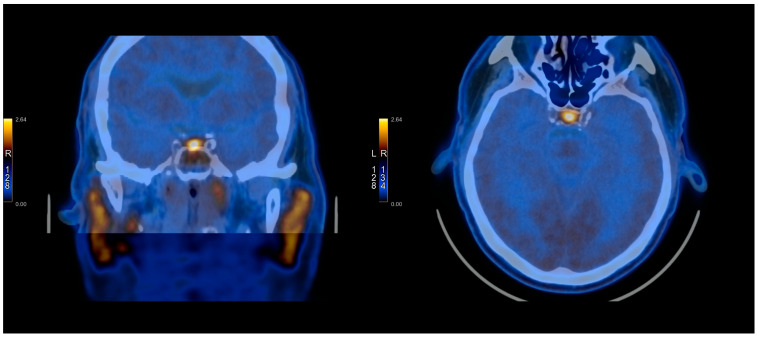
[^11^C]C-MET PET/CT study—False negative (Patient No. 5).

**Figure 5 cancers-17-03147-f005:**
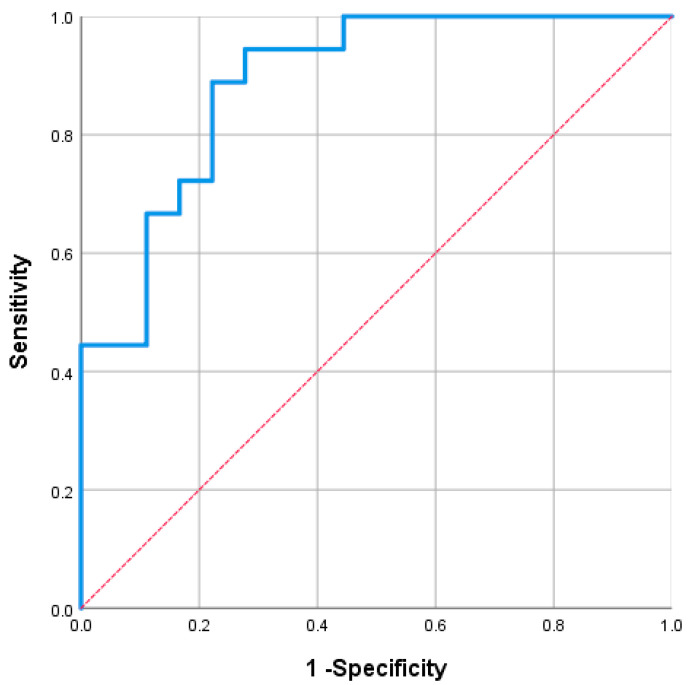
ROC curve—for SUV_max_ values in detecting pituitary adenoma. For ROC analysis, data of 38 patients were used (18 patients with histologically confirmed CD—“final study group”, and 18 patients with no pituitary abnormality—“control group”).

**Table 1 cancers-17-03147-t001:** Sensitivity, specificity, positive predictive value (PPV) and negative predictive value (NPV).

*n* = 21	Sensitivity	Specificity	PPV	NPV
MRI	59	75	91	70
[^11^C]C-MET PET/CT	23	13	30	91
BIPSS	100	100	100	0

MRI—magnetic resonance imaging, BIPSS—bilateral inferior petrosal sinus sampling.

**Table 2 cancers-17-03147-t002:** Clinical data of the “final” study group.

*n* = 18	Unit	M	SD
Age	y.o.	50.6	14.5
Activity	MBq	541.7	40.7
SUV_max_	-	3.74	0.87
Volume	mL	5.2	2.7
Morning cortisol concentration	µg/dL	27.4	20.6
Night cortisol concentration	µg/dL	20.0	12.9
ODST	µg/dL	15.5	7.5
Increase in ACTH on the adenoma side	x	7.1	5.4
Initial ACTH IPSS/peripheral	x	9.1	6.5
		Med.	IQR
Peak ACTH IPSS/peripheral	x	21.2	22.8
DHEAS	µg/dL	321.0	345.0
UFC	µg/24 h	240.7	347.2
ACTH	pg/mL	66.4	76.3

SUV_max_—maximum standardized uptake value; ODST—overnight dexamethasone suppression test (1 mg), ACTH—adrenocorticotropic hormone, DHEAS—dehydroepiandrosterone sulfate, UFC—urine free cortisol (24 h), y.o.—years old; M—mean, SD—standard deviation.

## Data Availability

The datasets used and/or analyzed during the current study are available from the corresponding author on reasonable request.

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
