# Peer review of "Comparison of Localization Methods in Cushing Disease—Could [11C]C-Methionine PET/CT Replace MRI or BIPSS?"

_cancers, 2025, doi:10.3390/cancers17193147_

Round 1
Reviewer 1 Report
Comments and Suggestions for Authors
- There are several studies on similar topic with contrasting results, the results presented in this paper need to compared with earlier published studies (just a few of them are hereunder)
"Wang, Z., Feng, Z., Zhu, D. et al. Clinical application of combination [11C]C-methionine and [13N]N-ammonia PET/CT in recurrent functional pituitary adenomas with negative MRI or [18F]F-FDG PET/CT. BMC Endocr Disord 24, 19 (2024). https://doi.org/10.1186/s12902-024-01543-2"
" Imaging Spectrum and Pitfalls of 11C-Methionine Positron Emission Tomography in a Series of Patients with Intracranial Lesions"
"11C-methionine-PET for differentiating recurrent brain tumor from radiation necrosis: radiomics approach with random forest classifier"
"Diagnostic Value of 11C-Methionine PET-CT Imaging in Persistent or Recurrent Cushing Disease After Surgery "
"Value of 11C-methionine PET in imaging brain tumours and metastases"
- Operational procedures must be presented with complete details. The manuscript states that the PET/CT images were evaluated by two nuclear medicine specialists and confirmed by a third.
- However, it does not state that these reviewers were blinded to the results of the BIPSS, MRI, or the clinical diagnosis.
- The control group was not part of the original prospective design. It was added retrospectively from a different study (patients with tertiary hyperparathyroidism).
- The manuscript states the control group had "pituitary adenomas or any pituitary abnormalities excluded," but the method of this exclusion (e.g was a pituitary MRI performed on all controls?) is not detailed.
- The protocol was: Day 1: PET/CT, Day 2: BIPSS, Day 3: MRI. BIPSS involves the injection of contrast media and a stimulus (desmopressin).
- Confusing notation for [11C]Methionine or [11C]MET repeatedly
- Statistical analyses description is very unclear stating "Pearson or Spearman tests were used for correlation analysis" but does not specify the exact correlations tested or how the choice between Pearson and Spearman was made (likely based on normality, but this should be stated).
- BIPSS protocol is described in sufficient details, while the PET/CT acquisition parameters were described very casually.
- The caption for Figure 5 is "ROC curve for SUVmax values in detecting pituitary adenoma." However, the results section states this ROC analysis was performed on the retrospective comparison between the final study group (n=18) and the control group (n=18), not on the primary prospective cohort of 21 patients
Author Response
Dear Reviewer 1
We would like to express our sincere gratitude for your valuable suggestions. Following your recommendations, we have addressed all the queries. With these improvements, we hope the paper now meets your expectations.
Best regards
Adam Durma, MD PhD
- There are several studies on similar topic with contrasting results, the results presented in this paper need to compared with earlier published studies (just a few of them are hereunder)
"Wang, Z., Feng, Z., Zhu, D. et al. Clinical application of combination [11C]C-methionine and [13N]N-ammonia PET/CT in recurrent functional pituitary adenomas with negative MRI or [18F]F-FDG PET/CT. BMC Endocr Disord 24, 19 (2024). https://doi.org/10.1186/s12902-024-01543-2"
" Imaging Spectrum and Pitfalls of 11C-Methionine Positron Emission Tomography in a Series of Patients with Intracranial Lesions"
"11C-methionine-PET for differentiating recurrent brain tumor from radiation necrosis: radiomics approach with random forest classifier"
"Diagnostic Value of 11C-Methionine PET-CT Imaging in Persistent or Recurrent Cushing Disease After Surgery "
"Value of 11C-methionine PET in imaging brain tumours and metastases"
Text was updated. We added some additional paragraphs, and suggested references were included.
2. Operational procedures must be presented with complete details. The manuscript states that the PET/CT images were evaluated by two nuclear medicine specialists and confirmed by a third.
3. However, it does not state that these reviewers were blinded to the results of the BIPSS, MRI, or the clinical diagnosis.
The operational procedures were updated. Two of the first nuclear medicine specialist were blinded to all of the patients results, and they were not part of the study team/ article authors. The third one (M.K.) confirmed all the results while being non-blinded.
4. The control group was not part of the original prospective design. It was added retrospectively from a different study (patients with tertiary hyperparathyroidism).
The control group was indeed not the part of the original prospective design. It was added retrospectively in order to increase the value of the paper, and to perform ROC analysis.
5. The manuscript states the control group had "pituitary adenomas or any pituitary abnormalities excluded," but the method of this exclusion (e.g was a pituitary MRI performed on all controls?) is not detailed.
Text was updated. We excluded patients with previously known pituitary lesions (adenomas or Rathke’s Cleft Cysts), and those 18 patients had functional pituitary adenoma excluded by laboratory tests (PRL, TSH, LH, FSH, IGF-1, ODT). We did not perform additional MRI in “control group”.
6. The protocol was: Day 1: PET/CT, Day 2: BIPSS, Day 3: MRI. BIPSS involves the injection of contrast media and a stimulus (desmopressin).
Yes, that is true. The BIPSS uses iodine contrast media to localize catheters and 10mg of desmopressin was used for stimulation tests (as CRH was unavailable).
7. Confusing notation for [11C]Methionine or [11C]MET repeatedly
Text was updated. We unified to [11C]C-MET
8. Statistical analyses description is very unclear stating "Pearson or Spearman tests were used for correlation analysis" but does not specify the exact correlations tested or how the choice between Pearson and Spearman was made (likely based on normality, but this should be stated).
Statistical description was updated.
9. BIPSS protocol is described in sufficient details, while the PET/CT acquisition parameters were described very casually.
PET/CT details were updated.
10. The caption for Figure 5 is "ROC curve for SUVmax values in detecting pituitary adenoma." However, the results section states this ROC analysis was performed on the retrospective comparison between the final study group (n=18) and the control group (n=18), not on the primary prospective cohort of 21 patients
We have corrected the text to avoid misleads. During analysis we considered additional control group could improve statistical analysis and make results more reliable.
Reviewer 2 Report
Comments and Suggestions for Authors
The article “Comparison of localization methods in Cushing Disease – could [¹¹C]C-methionine PET/CT replace MRI or BIPSS?” presents a prospective study that included 21 patients with Cushing Disease (CD) who underwent BIPSS, MRI, and [¹¹C]C-MET PET/CT of the pituitary. The manuscript requires major improvements in both study design and data presentation. Please address the following comments for substantial revision of the manuscript:
- Please introduce other PET imaging agents used in Cushing Disease (CD) in the introduction and clearly state the hypothesis for selecting [¹¹C]C-MET in this study.
- Please provide a detailed synthesis protocol for [¹¹C]C-MET in Section 2.4, including all relevant parameters (e.g., radiochemical purity [RCP], radiochemical yield [RCY], and molar activity).
- Figures 1–4 require additional information such as the location of the regions of interest (ROIs), PET acquisition time window, and standardized PET scale bars.
- Please incorporate the clinical data for MRI (images) and BIPSS in the Results section for the patients involved in this study.
- Please label each image with the corresponding patient identifier to enable correlation with the results.
Author Response
Dear Reviewer 2
We would like to thank you for the revision and your valuable suggestions. Following your recommendations, we have addressed all the queries. With these improvements, we hope the paper now meets your expectations.
Best regards
Adam Durma, MD PhD
- Please introduce other PET imaging agents used in Cushing Disease (CD) in the introduction and clearly state the hypothesis for selecting [¹¹C]C-MET in this study.
Text was updated, and additional agents were introduced and described.
2. Please provide a detailed synthesis protocol for [¹¹C]C-MET in Section 2.4, including all relevant parameters (e.g., radiochemical purity [RCP], radiochemical yield [RCY], and molar activity).
Synthesis protocol paragraph was added.
3. Figures 1–4 require additional information such as the location of the regions of interest (ROIs), PET acquisition time window, and standardized PET scale bars.
Figures were restored to primary status due to suggestion – Supplementary Figures with all details were added at the end of the article.
4. Please incorporate the clinical data for MRI (images) and BIPSS in the Results section for the patients involved in this study.
Results paragraph was corrected. Clinical data for BIPSS are presented in Table 2.
5. Please label each image with the corresponding patient identifier to enable correlation with the results.
Images were labeled with patient’s identifiers (Patient No. X).
Round 2
Reviewer 1 Report
Comments and Suggestions for Authors
The authors have addressed my comments. and I feel this manuscript may be recommended for acceptance
Reviewer 2 Report
Comments and Suggestions for Authors
The updated manuscript can be accepted for publication.